# A fragment-based drug discovery developed on ciclopirox for inhibition of Hepatitis B virus core protein: An in silico study

**Alireza Mohebbi**[1,2]*, **Touba Ghorbanzadeh**[3◉], **Shabnam Naderifar**[4◉], **Fattaneh Khalaj**[5,6◉], **Fatemeh Sana Askari**[2‡], **Ali Salehnia Sammak**[7‡]

**1** Department of Virology, School of Medicine, Iran University of Medical Sciences, Tehran, Iran, **2** Vista Aria Rena Gene Inc., Golestan Province, Gorgan, Iran, **3** Department of Microbiology, Tehran North Branch, Islamic Azad University of Tehran, Tehran, Iran, **4** Department of Biotechnology, Brandenburg Technology University, Senftenberg, Germany, **5** Liver and Pancreatobiliary Diseases Research Center, Shariati Hospital, Tehran University of Medical Sciences, Tehran, Iran, **6** Digestive Diseases Research Center, Shariati Hospital, Tehran University of Medical Sciences, Tehran, Iran, **7** Department of Microbiology, Rasht Branch, Islamic Azad University, Rasht, Iran

◉ These authors contributed equally to this work.
‡ FSA and ASS also contributed equally to this work.
* Alirezaa2s@gmail.com

**Data Availability Statement:** All relevant data are within the paper and its Supporting Information files.

## Abstract

The Hepatitis B virus (HBV) core protein is an attractive target for preventing capsid assembly and viral replication. Drug repurposing strategies have introduced several drugs targeting HBV core protein. This study used a fragment-based drug discovery (FBDD) approach to reconstruct a repurposed core protein inhibitor to some novel antiviral derivatives. Auto Core Fragment in silico Screening (ACFIS) server was used for deconstruction-reconstruction of Ciclopirox in complex with HBV core protein. The Ciclopirox derivatives were ranked based on their free energy of binding ($\Delta$GB). A quantitative structure affinity relationship (QSAR) was established on the Ciclopirox derivatives. The model was validated by a Ciclopirox-property-matched decoy set. A principal component analysis (PCA) was also assessed to define the relationship of the predictive variable of the QSAR model. 24-derivatives with a $\Delta$GB (-16.56±1.46 Kcal.mol$^{-1}$) more than Ciclopirox was highlighted. A QSAR model with a predictive power of 88.99% (F-statistics = 9025.78, corrected df(25), Pr > F = 0.0001) was developed by four predictive descriptors (ATS1p, nCs, Hy, F08[C-C]). The model validation showed no predictive power for the decoy set ($Q^2$ = 0). No significant correlation was observed between predictors. By directly attaching to the core protein carboxyl-terminal domain, Ciclopirox derivatives may be able to suppress HBV virus assembly and subsequent viral replication inhibition. Hydrophobic residue Phe23 is a critical amino acid in the ligand binding domain. These ligands share the same physicochemical properties that lead to the development of a robust QSAR mode. The same strategy may also be used for future drug discovery of viral inhibitors.

**Funding:** The author(s) received no specific funding for this work.

**Competing interests:** The authors have declared that no competing interests exist.

## Introduction

The Hepatitis B virus (HBV) is the cause of chronic hepatitis B (CHB), liver cirrhosis, and hepatocellular carcinoma (HCC). Despite the recombinant vaccine, more than 300 million people worldwide have CHB, and more than 300,000 people die annually from complications of HBV infection, such as HCC [1, 2].

Approved treatments for HBV infection include nucleos(t)ide analogs (NUCs) that inhibit the reverse transcription activity of P protein, but in most cases, discontinuing therapy leads to viral replication relapse and the emergence of drug-resistant strains. Another approved treatment stimulates innate immunity by interferon-alpha (IFN-α). Long-term use of IFN-α has a low success rate and is associated with severe side effects in more than 30% of cases. In addition, viral particles empty of the genome are produced by S proteins, which can be active in modulating immune responses during replication. Also, the presence of cccDNA in liver cells is a factor in persistent virus replication activity [3]. Various research results show that it is impossible to get complete treatment except by reversing the positive serological responses to S antigen and abolishing reservoir cccDNA. Recent studies show that core proteins play an essential role in cccDNA formation and virus replication [4].

HBV is an enveloped virus with an incompletely circled double-stranded DNA (rcDNA) inside the virus capsid. After initial binding to heparan sulfate proteoglycan (HSPG) on the cell surface, the virus binds to its primary receptor called sodium-taurocholate co-transporting polypeptide (NTCP) through the large surface protein (LHBsAg) preS1 domain and enters the cytoplasm with the help of the epidermal growth factor receptor (EGFR) [5, 6]. After uncoating, the virus takes its genome into the nucleus through the nuclear transfer signal on its nucleocapsid proteins. In the nucleus, the rcDNA genome of the virus turns it into a complete double-stranded (cccDNA) by the host's DNA repair enzymes. cccDNA plays a vital role in the persistence of viral infection and serves as a template for transcribing the viral genome. Seven proteins are made from 4 copies of viral mRNA, including three surface proteins (S, M, and L), polymerase (P), core protein or capsid (C), accessory protein (e), and transactivator X protein, respectively. The 3.5 kb version (pgRNA) and the P protein are taken into the capsid by the nuclear proteins of the virus, where reverse transcription and rcDNA production occur in new viral particles. The life cycle of HBV and current knowledge of its different stages of the viral life cycle have been used as a target for treatment so far [7].

The HBV core is a structural protein that plays important roles in the viral life cycle, including transmission and protection of viral genetic information, viral replication, and virion assembly. Depending on the genotype, the core protein contains 183 to 185 amino acids, forming a capsid composed of 120 copies of core protein dimers. The core protein contains three regions, including an amino(N)-terminal domain (NTD; amino acids 1–140), the carboxyl (C)-terminal domain (CTD; amino acids 150–183), and a flexible linker containing nine amino acids (amino acids 141–149) linking the N and C domains [8]. The NTD plays a central role in the capsid assembly, while the arginine-rich CTD is involved in pgRNA packaging [9]. In addition, the core protein contains nuclear localization and nuclear exit signals (NL/ES) in the arginine repeat region of the CTD, which enable the core protein to translocate between the nucleus and the cytoplasm efficiently. Evidence suggests that the CTD and linker are essential for capsid assembly and viral genome replication [10].

Today, clinical studies are in phases 1 and 2, where the core protein is targeted. The compounds with the inhibition activity of core protein are called capsid assembly modulators (CAM). CAMs can be classified into two categories based on the post-treatment capsid states. Class I molecules, including heteroaryl-dihydro pyrimidines (HAP) such as GLS4, RO7049389, Bay 41–4109, and HAP_R10, which cause the formation of polymers on the

capsid or large aggregates of core proteins. While class II molecules, including phenylpropena-mides (PPAs) such as AT130 and sulfamoylbenzamides (SBAs) such as AB-423 and NVR 3–778, cause the formation of empty capsids without pgRNA. Both CAMs bind to a pocket in the dimeric polymer-forming region of core proteins and induce error-prone core protein assembly, ultimately preventing the formation of infectious virions [11–18]. Recently, Kang et al. have shown that an FDA-approved drug, Ciclopirox, can directly bind and occupy the hydrophobic pocket of HBV capsid protein CTD and hampers the formation of HBV capsid polymers. Ciclopirox is a hydroxypyridone antifungal agent with therapeutic functions in cancer [19, 20], ischemic stroke [21], and oral candidiasis [22]. Further studies have also shown that the ethanolamine salt of Ciclopirox has antiviral properties against human Herpes Simplex Viruses (HSV-1 & HSV-2) [23, 24]. Therefore, reconstruction of Ciclopirox, a low molecular weight chemical compound with 32 atoms, as a chemical starting point, into more potent drug-like leads with higher affinity to the HBV core protein is a promising approach. In this regard, the present study aims to expand the chemical space for identifying potent small molecules by deconstructing Ciclopirox into its fragments and expanding fragments into novel potent HBV core protein inhibitors.

## Materials and methods

### Data gathering of HBcAg and Ciclopirox

The crystallographic structure of HBcAg of HBV genotype D subtype adw was obtained from a protein data bank (PDB) with the PDB ID of 6j10 and 2.30 Å resolution [25]. The structure of HBcAg was cleaned from water molecules and other non-standard fragments, as previously shown [26, 27]. Furthermore, the chemical structure of Ciclopirox in complex with the HBcAg was also extracted from the same.pdb file. The complex.pdb file of HBcAg and Ciclopirox was used for fragment-based drug development.

### Fragment-based drug discovery

Auto Core Fragment in silico Screening (ACFIS) server [28] was used to expand Ciclopirox's chemical space and identify novel Ciclopirox-scaffold-based chemical compounds. This also helped to improve the number of anti-HBcAg hits with increased efficiency. PARA_GEN, CORE_GEN, and CAND GEN modules were used to generate core fragment structure from the Ciclopirox molecule using fragment deconstruction analysis and to perform in silico screening by growing fragments to the junction of the core fragment. The free energy of binding, $\Delta G_{bind}$ ($\Delta GB$) [29], of Ciclopirox within HBcAg was used as the cut-off for sampling novel generated compounds for further analysis. Here,$\Delta GB$ was used for ranking Ciclopirox derivatives, and it includes binding energy, solvation entropy, and conformational entropy [28, 30].

### ADMET prediction

Physicochemical properties, Lipophilicity, Water Solubility, Pharmacokinetics, and Druglikeness of the compounds were assessed by the SwissADME server [31]. Accordingly, chemical structures' smile files were used to assess the ADME properties of the compounds. The toxicity of the chemicals was also assessed by the toxicity estimation software tool (T.E.S.T) [32]. In this regard, the nearest neighbor method was used to predict *Fathead minnow* $LC_{50}$ (96 hr), *Daphnia magna* $LC_{50}$ (48 hr), *Tetrahymena pyriformis* $IGC_{50}$ (48 hr), Oral rat $LC_{50}$, Bioaccumulation factor, and Ames mutagenicity as previously shown [33].

## Quantitative structure affinity relationship (QSAR)

A model was developed by investigating the relationship of the chemical descriptors with the hits' affinity to HBcAg protein. Accordingly, a stepwise multiple linear regression (MLR) was utilized to select a set of descriptors with no overfitting penalty to see the correlation of chemical fragment descriptors with the affinity of the Ciclopirox-core fragment-derivatives, as defined before [34]. For this purpose, 2D descriptors, including information indices, edge adjacency indices, topological charge indices, topological descriptors, connectivity indices, 2D autocorrelations, 2D binary fingerprints, and 2D frequency fingerprints were predicted with DRAGON 5.5 software [35]. Descriptors with zero variance were removed. The MLR model was developed by MS-Excel 2019. Accordingly, Ciclopirox and its derivatives were flagged (n = 25) as the training set.

Additionally, the model was validated with 614 Ciclopirox-property-matched decoys as the test set with no anti-HBV reports (inactive). The decoys were generated by the Directory of Useful Decoys- Enhanced (DUD-E) server [36, 37]. The affinity of the training and test sets of small molecules were predicted by Autodock Vina [38] as described before [27, 33] and used as the dependent variable. Descriptors with a probability of entry of $\leq 0.01$ were selected as independent variables, and the model was developed with a confidence interval (CI) of 95%. The goodness of fit statistics was assessed to investigate the model quality. Principal component analysis (PCA) was also performed to investigate the association between predictors and their impact on the affinity of ligands toward HBcAg.

## Results

### HBcAg and Ciclopirox

The cleaned complex structure of HBcAg and Ciclopirox was used for fragment-based drug discovery (FBDD). As shown in Fig 1, Ciclopirox was deconstructed into its core fragments. The core fragments were extended to the novel, efficient ligands. The novelty of the ligands was checked in the PubChem database [39]. Forty-one ligands were generated based on the Ciclopirox core fragments (data are not shown). ΔGB of the Ciclopirox (-14.79 Kcal/mol) was used as the binding affinity cut-off and the positive control to investigate the efficiency of the ligands. Accordingly, the ligands with lower affinities than that of Ciclopirox were discarded. Accordingly, 24 ligands (Table 1) were selected for ADMET prediction and for establishing a robust SAR model. The mean ΔGB of developed ligands was -16.56±1.46 Kcal/mol. Ligand118 had the highest affinity (-19.418) to HBcAg relative to other small molecules. The hits were assessed for their ADMET properties (Table 2). As shown in Fig 2, the binding site of the ligands with the highest affinity to the HBV core protein at the NTD comprises F23, F24, Y118, F122, P25, W102, and L19 atoms were in close interactions with the Lig118. Accordingly, 91 close contacts (-0.06±0.27 Å) were between Lig118 and the mentioned capsid protein residues.

### ADMET prediction of Ciclopirox derivatives

The results demonstrated soluble drug-likeness properties of the 24 Ciclopirox derivative ligands. The mean molecular weight (MW) of the ligands was 209.80±33.57 Kda. The toxicity results demonstrated positive predicted Ames toxicity among 6/15 ligands with validated results (Table 2). Only non-toxicant ligands with negative Ames toxicity were used for MDS and Ciclopirox (Table 3).

### The structure affinity relationship mode

Q SAR equation was established to investigate the prediction ability of 25 (24 ligands plus ciclopirox) ligands' 2D descriptors with their ΔGB values. Accordingly, 2489 descriptors were

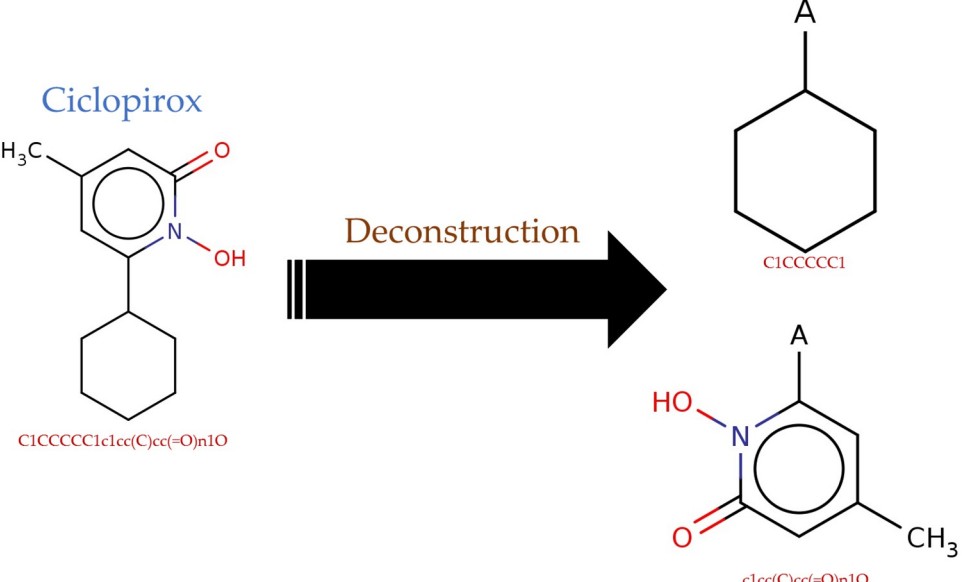

**Fig 1. CORE_GEN deconstruction of Ciclopirox in two core fragments with -3.11 Kcal/mol and -8.52 Kcal/mol affinities to HBcAg, respectively.**

predicted for each ligand, of which 1186 constant, 48 near-constant, and 47 correlated (Pearson r = 1) variables were excluded. A total of 1208 descriptors remained for QSAR model establishment. Table 4 shows the summary of the descriptors selected for the QSAR model (the raw data is also available in the S1 File). As shown, the model with four variable (ATS1p, nCs, Hy, and F08[C-C]) was the best predictive model (adjusted $R^2$ = 0.8799, F-statistics = 9025.78, corrected df(25), Pr > F = 0.0001).

The goodness of fit statistics was used to examine the power of descriptors in the model and test-set and overfitting. It was found that 99.93% of the variance of the model was explained by ATS1p, nCs, Hy, and F08[C-C] (Adjuster $R^2$ = 0.9993, MSE = 0.0358). The model fitness was assessed by the square root of the variance of the residuals (RMSE] = 0.1892 in the model versus 1.1383 for the test set), Chi-square $Q^2$ (0.9999 versus 0.0000 for the training set and test set, respectively) and PRESS statistics (0.9616). Further fitness of the model is provided in the residual plot (Fig 3A). Furthermore, Cook's distance (Fig 3B) of 25 training sets was less than 0.16.

The association of the four descriptors in the model was assessed by principal component analysis (PCA). As shown in the Scree plot (Fig 4A), 77.32% of the cumulative variance was explained by the first two principal components (PCs). A factor, the number of hydrogen donors (nHDon), was used as a dummy (indicator) variable to distinguish the multicollinearity of the main predictors. Factor loadings showed a strong contribution to the indicator (0.8578), Hy (0.8482), ATS1p (-0.7086), and F08[C-C] (-0.6497) to the PC1. Furthermore, F08 [C-C] (0.6333) and nCs (0.6485) had a strong contribution to PC2, and ATS1p (0.3786) and Hy (0.4936) had a moderate contribution to PC2. The multicollinearity of variables in the model is shown in the biplot (Fig 4B). Accordingly, ATS1p was positively correlated with F08 [C-C] (0.6523). Hy and the indicator also had a high correlation (0.9856). There was no correlation between Hy with nCs and F08[C-C] and nCs with ATS1p. Furthermore, the vectors Hy (29.5957) and the indicator (30.2714) contributed more to PC1. Additionally, nCs (29.3121) and F08[C-C] (27.9486) were more contributors to PC2. Furthermore, the ligands' distribution to the predictors is shown in Fig 4B. Accordingly, ligand272 cutpoints were far off the

**Table 1. Affinity (kcal/mol) for the different Ciclopirox derivatives.**

| No. | ΔH (PB)[‡] | ΔH (GB)[§] | -TΔS[¥] | ΔG (PB)[†] | ΔG (GB)[₵] | Smiles |
|---|---|---|---|---|---|---|
| ligand118 | -22.34 | -24.91 | 5.492 | -16.848 | -19.418 | C1CCCCC1[C@H]1[N@H+](C)CCCC1 |
| ligand122 | -22.13 | -26.35 | 7.13 | -15 | -19.22 | C1CCCCC1C[NH+]1CCCCC1 |
| ligand164 | -23.04 | -26.8 | 7.981 | -15.059 | -18.819 | C1CCCCC1c1cc2c(oc (= O)cc2O)cc1 |
| ligand272 | -22.48 | -25.68 | 7.062 | -15.418 | -18.618 | C1CCCCC1C (= O)O[C@@H]1Cc2c(CC1)cc(cc2)F |
| ligand80 | -22.91 | -26.77 | 8.693 | -14.217 | -18.077 | C1CCCCC1c1ccc(cc1)N (= O) = O |
| ligand163 | -20.12 | -24.87 | 7.064 | -13.056 | -17.806 | C1CCCCC1c1c2c(oc (= O)cc2O)ccc1 |
| ligand251 | -20.03 | -24.06 | 6.471 | -13.559 | -17.589 | C1CCCCC1c1c2[nH]cnc2ccc1 |
| ligand162 | -22.46 | -25.41 | 8.042 | -14.418 | -17.368 | C1CCCCC1c1c (= O)oc2c(c1O)cccc2 |
| ligand167 | -20.19 | -24.34 | 7.64 | -12.55 | -16.7 | C1CCCCC1Oc1cc (= O)oc2c1cccc2 |
| ligand81 | -19.12 | -23.32 | 6.628 | -12.492 | -16.692 | C1CCCCC1c1cccc(c1)N (= O) = O |
| ligand248 | -21.58 | -24.11 | 7.432 | -14.148 | -16.678 | C1CCCCC1c1c2c([nH]cn2)ccc1 |
| ligand79 | -18.96 | -23.24 | 6.585 | -12.375 | -16.655 | C1CCCCC1c1cccc(c1)N (= O) = O |
| ligand226 | -19.44 | -22.17 | 6.172 | -13.268 | -15.998 | C1CCCCC1Cc1cc(ccc1)C |
| ligand250 | -21.39 | -24.12 | 8.262 | -13.128 | -15.858 | C1CCCCC1c1ccc2c([nH]cn2)c1 |
| ligand229 | -18.93 | -22.25 | 6.507 | -12.423 | -15.743 | C1CCCCC1c1cc(cc(C)c1)C |
| ligand105 | -20.69 | -23.22 | 7.495 | -13.195 | -15.725 | C1CCCCC1c1c(cccc1)N |
| ligand280 | -22.94 | -25.97 | 10.366 | -12.574 | -15.604 | C1CCCCC1c1c2C[C@H](CCc2cc(c1)F)OC = O |
| ligand274 | -17.68 | -22.09 | 6.801 | -10.879 | -15.289 | C1CCCCC1[C@H]1[C@H](CCc2c1ccc(c2)F)OC = O |
| ligand230 | -19.84 | -22.6 | 7.529 | -12.311 | -15.071 | C1CCCCC1c1c(C)cc(cc1)C |
| ligand228 | -19.65 | -22.53 | 7.53 | -12.12 | -15 | C1CCCCC1c1c(cc(C)cc1)C |
| ligand223 | -18.9 | -21.7 | 6.77 | -12.13 | -14.93 | C1CCCCC1[C@H]1CCC[NH2+]C1 |
| ligand36 | -18.4 | -20.9 | 5.984 | -12.416 | -14.916 | C1CCCCC1c1cccc(C)c1 |
| ligand112 | -19.29 | -22.36 | 7.526 | -11.764 | -14.834 | C1CCCCC1c1ccc(cc1)F |
| ligand282 | -21.37 | -22.92 | 8.125 | -13.245 | -14.795 | C1CCCCC1C[C@H]1C[C@H](CCO1)O |

‡ Poisson–Boltzmann enthalpy

§ Conformational enthalpy upon binding

¥ Conformational entropy upon binding

† Poisson–Boltzmann free energy

₵ Free energy of binding

predictors, representing the higher values of F08[C-C], ATS1p, and nCs. Ligand163 was close to the origin, representing the lower factor values. Ligand272 showed a high distance with ligand223, Ciclopirox, and ligand105. It is worth to mention the Autodock vina was performed on both training and test sets to prevent bias toward SAR modeling.

## Discussion

Chronic HBV infection is a critical public health issue due to irresponsiveness to viral NUCs and immunotherapies. There are currently approved recombinant vaccine and NUCs, including lamivudine (3TC), adefovir disoproxil (ADF), entecavir (ETV), telbivudine (LdT), tenofovir disoproxil fumarate (TDF), and tenofovir alafenamide (TAF). Furthermore, pegylated interferon alpha (PEG-IFN -α) in complementation with NUCs is approved for HBV therapy. However, such interventions' efficiency is inadequate owing to the residual hepatocytes carrying cccDNA, HBsAg production, drug-resistance mutation, and vaccine-scape mutants. Promising strategies targeting host receptor and viral HBV core protein are now undergoing clinical trials. However, the drug repurposing approach has gained much attention partly because of the COVID-19 pandemic [40]. Accordingly, different studies have been performed in vitro

**Table 2. Druglikeness and descriptor prediction of Ciclopirox derivatives.**

| Molecules | Canonical SMILES | Formula | MW (Da) | Rotatable bonds | H-bond acceptors | H-bond donors | TPSA | XLOGP3 | Solubility (mg/ml) | Solubility (mol/l) | ESOL Class | GI absorption | Bioavailability Score | Druglikeness | | | | |
|---|---|---|---|---|---|---|---|---|---|---|---|---|---|---|---|---|---|---|
| | | | | | | | | | | | | | | Lipinski | Ghose | Veber | Egan | Muegge |
| ligand118 | C[NH+]1CCCC[C@H]1C1CCCCC1 | C12H24N | 182.33 | 1 | 0 | 1 | 4.44 | 3.55 | 1.32E-01 | 7.23E-04 | Soluble | Low | 0.55 | + | + | + | + | - |
| ligand122 | C1CCC(CC1)C[NH+]1CCCCC1 | C12H24N | 182.33 | 2 | 0 | 1 | 4.44 | 3.48 | 1.70E-01 | 9.31E-04 | Soluble | Low | 0.55 | + | + | + | + | - |
| ligand164 | O=c1cc(O)c2c(o1)ccc(c2)C1CCCCC1 | C15H16O3 | 244.29 | 1 | 3 | 1 | 50.44 | 3.81 | 1.94E-02 | 7.94E-05 | Moderately soluble | High | 0.55 | + | + | + | + | + |
| ligand272 | O=C(C1CCCCC1)O[C@H]1CC2c(C1)ccc(c2)F | C17H21FO2 | 276.35 | 3 | 3 | 0 | 26.3 | 4.71 | 7.89E-03 | 2.85E-05 | Moderately soluble | High | 0.55 | + | + | + | + | + |
| ligand80 | O=N(=O)c1ccc(cc1)C1CCCCC1 | C12H15NO2 | 205.25 | 2 | 2 | 0 | 45.82 | 4.16 | 2.60E-02 | 1.27E-04 | Soluble | High | 0.55 | + | + | + | + | + |
| ligand163 | O=c1cc(O)c2(o1)ccc2C1CCCCC1 | C15H16O3 | 244.29 | 1 | 3 | 1 | 50.44 | 3.81 | 1.94E-02 | 7.94E-05 | Moderately soluble | High | 0.55 | + | + | + | + | + |
| ligand251 | C1CCC(CC1)c1cccc2c1[nH]cn2 | C13H16N2 | 200.28 | 1 | 1 | 1 | 28.68 | 4.02 | 2.04E-02 | 1.02E-04 | Soluble | High | 0.55 | + | + | + | + | + |
| ligand162 | O=c1oc2ccccc2c(c1C1CCCCC1)O | C15H16O3 | 244.29 | 1 | 3 | 1 | 50.44 | 3.82 | 1.91E-02 | 7.83E-05 | Moderately soluble | High | 0.55 | + | + | + | + | + |
| ligand167 | O=c1cc(OC2CCCCC2)c2c(o1)cccc2 | C15H16O3 | 244.29 | 2 | 3 | 0 | 39.44 | 3.41 | 4.04E-02 | 1.65E-04 | Soluble | High | 0.55 | + | + | + | + | + |
| ligand81 | O=N(=O)c1ccc(c1)C1CCCCC1 | C12H15NO2 | 205.25 | 2 | 2 | 0 | 45.82 | 4.62 | 1.33E-02 | 6.50E-05 | Moderately soluble | High | 0.55 | + | + | + | + | + |
| ligand248 | C1CCC(CC1)c1ccc2c1nc[nH]2 | C13H16N2 | 200.28 | 1 | 1 | 1 | 28.68 | 4.02 | 2.04E-02 | 1.02E-04 | Soluble | High | 0.55 | + | + | + | + | + |
| ligand79 | O=N(=O)c1ccc(cc1)C1CCCCC1 | C12H15NO2 | 205.25 | 2 | 2 | 0 | 45.82 | 4.62 | 1.33E-02 | 6.50E-05 | Moderately soluble | High | 0.55 | + | + | + | + | + |
| ligand226 | Cc1ccc(c1)CC1CCCCC1 | C14H20 | 188.31 | 2 | 0 | 0 | 0 | 5.29 | 5.62E-03 | 2.98E-05 | Moderately soluble | Low | 0.55 | + | + | + | + | - |
| ligand250 | C1CCC(CC1)c1ccc2c(c1)[nH]cn2 | C13H16N2 | 200.28 | 1 | 1 | 1 | 28.68 | 4.02 | 2.04E-02 | 1.02E-04 | Soluble | High | 0.55 | + | + | + | + | + |
| ligand229 | Cc1cc(cc1C)C1CCCCC1 | C14H20 | 188.31 | 1 | 0 | 0 | 0 | 5.19 | 5.58E-03 | 2.96E-05 | Moderately soluble | Low | 0.55 | + | + | + | + | - |
| ligand105 | Nc1ccc1C1CCCCC1 | C12H17N | 175.27 | 1 | 0 | 1 | 26.02 | 3.78 | 4.57E-02 | 2.61E-04 | Soluble | High | 0.55 | + | + | + | + | - |
| ligand280 | O=CO[C@H]1CCc2c(C1)cc(c2)F)C1CCCCC1 | C17H21FO2 | 276.35 | 3 | 3 | 0 | 26.3 | 5.18 | 3.99E-03 | 1.44E-05 | Moderately soluble | High | 0.55 | + | + | + | + | - |
| ligand274 | O=CO[C@H]1CCc2c([C@H]1C1CCCCC1)ccc(c2)F | C17H21FO2 | 276.35 | 3 | 3 | 0 | 26.3 | 5.18 | 3.99E-03 | 1.44E-05 | Moderately soluble | High | 0.55 | + | + | + | + | - |
| ligand230 | Cc1cc(c(c1)C)C1CCCCC1 | C14H20 | 188.31 | 1 | 0 | 0 | 0 | 5.19 | 5.58E-03 | 2.96E-05 | Moderately soluble | Low | 0.55 | + | + | + | + | - |
| ligand228 | Cc1cc(c(c1)C)C1CCCCC1 | C14H20 | 188.31 | 1 | 0 | 0 | 0 | 5.19 | 5.58E-03 | 2.96E-05 | Moderately soluble | Low | 0.55 | + | + | + | + | - |
| ligand223 | C1CCC(CC1)[C@H]1CCC[NH2+]C1 | C11H22N | 168.3 | 1 | 0 | 1 | 16.61 | 3.28 | 2.20E-01 | 1.31E-03 | Soluble | High | 0.55 | + | + | + | + | - |
| ligand36 | Cc1ccc(c1)C1CCCCC1 | C13H18 | 174.28 | 1 | 0 | 0 | 0 | 4.82 | 1.02E-02 | 5.85E-05 | Moderately soluble | Low | 0.55 | + | + | + | + | - |
| ligand112 | Fc1ccc(cc1)C1CCCCC1 | C12H15F | 178.25 | 1 | 1 | 0 | 0 | 4.43 | 1.74E-02 | 9.74E-05 | Moderately soluble | Low | 0.55 | + | + | + | + | - |
| ligand282 | O[C@H]1CCO[C@H](C1)CC1CCCCC1 | C12H22O2 | 198.3 | 2 | 2 | 1 | 29.46 | 3.11 | 2.52E-01 | 1.27E-03 | Soluble | High | 0.55 | + | + | + | + | - |

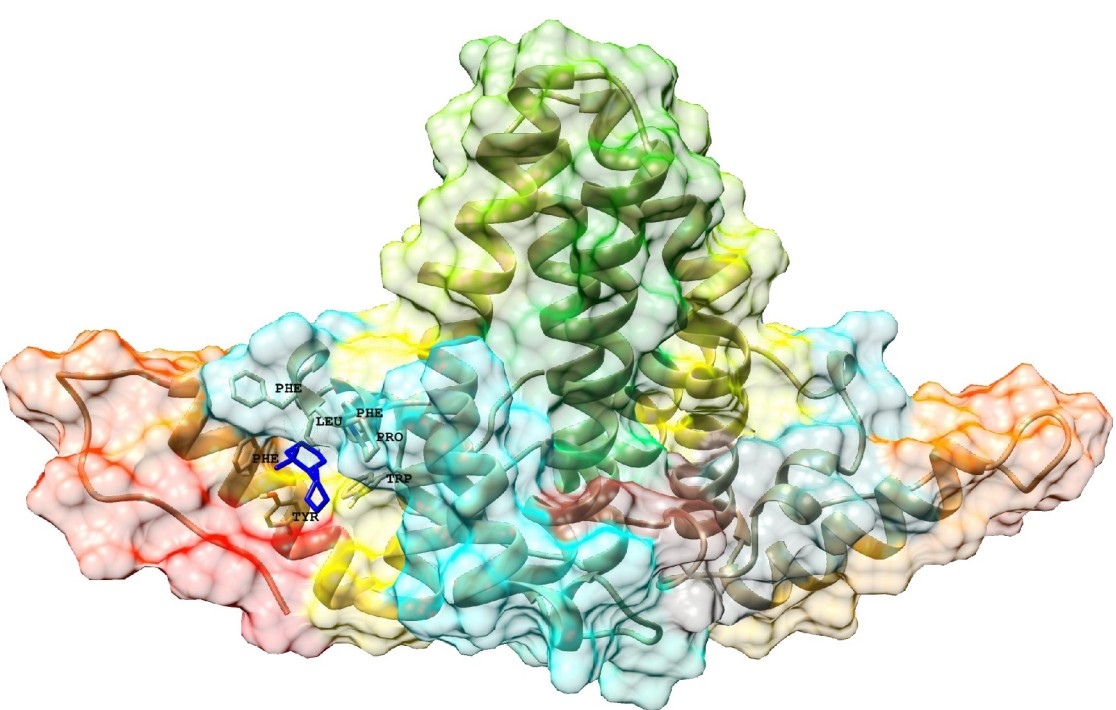

**Fig 2. Schematic representation of the Lig118 with the highest affinity to HBV core protein.** The residues F23 with 26 different interactions with Lig118. Further amino acid residues were F24, Y118, F122, P25, W102, and L19 with 6, 14, 13, 12, 10, and 10 interactions, respectively.

and in vivo studies to repurpose FDA-approved drugs for HBV therapy [25, 41, 42]. Therefore, in this computational-aided drug discovery study, we have used FBDD to highlight some potent HBV capsid assembly based on the physical shape of Ciclopirox.

The FBDD resulted in the development of potent HBV capsid inhibitors with a binding affinity score to the HBV capsid protein higher than that observed in Ciclopirox complexed with HBV core. Similarly, Zangi et al. demonstrated that the development of the core fragments of Ciclopirpx into novel derivatives might reduce HSV-1 and HSV-2 replication more than 1000-fold in vitro [24]. Therefore, the reconstruction of Ciclopirox enhances its therapeutic activities. Accordingly, 24 novel compounds were reconstructed with an average binding affinity of -16.56±1.46 Kcal.mol$^{-1}$, and all had higher affinities to the HBV capsid protein than Ciclopirox affinity to the receptor. As Ciclopirox is a small molecule, and in its crystallographic structure in complex with HBcAg it occupies a defined sub-binding pocket, the derivatives targeting the same binding site were small. The fragments, however, highlighted a more improved predicted binding affinities to the target protein. The CTD of HBV core protein is known to be involved in capsid dimerization. The CTD contains contributing amino acids in the Ciclopirox-receptor binding site, including Y118, F122, and W102. Here, it was found that the highlighted ligands with the highest binding affinities to the HBV capsid protein have the same binding site at the CTD, where F23 and S106 may play a significant role in close contact with the ligands. The reconstructed compounds had a similar binding site within CTD of HBV capsid protein. Therefore, it was supposed that it might be associated with their physicochemical composition. Therefore, the chemical descriptors were predicted for each compound for a quantitative structure-affinity relationship (QSAR) evaluation.

Based on our previous study [34], a protocol was developed for equating a QSAR model, based on which compounds with structural similarities can be applied in such a model. Here,

**Table 3. Toxicity prediction of Ciclopirox derivatives.**

| Molecules | Fathead minnow LC50 (96 hr) (mg/L) | Daphnia magna LC50 (mg/L) | T. pyriformis IGC50 (48 hr) (mg/L) | Oral rat LD50 (mg/kg) | Bioaccumulation factor (Predicted value) | Developmental Toxicity | Mutagenicity |
|---|---|---|---|---|---|---|---|
| ligand118 | 29.42 | N/A | 0.80 | 462.33 | N/A | NON-toxicant | N/A |
| ligand122 | 3.84 | N/A | 0.99 | 462.33 | N/A | NON-toxicant | N/A |
| ligand164 | 0.2 | 0.3 | 5.72 | 106.21 | 2290.87 | N/A | N/A |
| ligand272 | 0.29 | 1.14 | 3.58 | 862.49 | 178.51 | NON-toxicant | Negative |
| ligand80 | 0.86 | 4.15 | 3.37 | 749.27 | 107.40 | toxicant | Positive |
| ligand163 | N/A | N/A | N/A | N/A | N/A | N/A | N/A |
| ligand251 | 1.54 | 4.33 | 14.43 | 499.47 | 31.52 | toxicant | Positive |
| ligand162 | N/A | 3.41 | 7.32 | N/A | 2290.87 | N/A | N/A |
| ligand167 | N/A | 0.3 | 4.41 | N/A | 1933.24 | N/A | N/A |
| ligand81 | 1.12 | 3.67 | 3.10 | 727.24 | 80.18 | toxicant | Positive |
| ligand248 | 1.41 | 4.33 | 13.51 | 495.11 | 45.41 | NON-toxicant | Positive |
| ligand79 | 1.12 | 3.67 | 3.10 | 727.24 | 80.18 | toxicant | Positive |
| ligand226 | 1.22 | 0.75 | 1.98 | 2676.76 | 1201.26 | toxicant | Negative |
| ligand250 | 3.27 | 3.61 | 13.23 | 288.21 | 49.70 | toxicant | Positive |
| ligand229 | 0.88 | 0.97 | 2.39 | 3947.08 | 1279.67 | toxicant | Negative |
| ligand105 | 4.92 | 2.71 | 18.06 | 2504.85 | 47.44 | toxicant | Negative |
| ligand280 | N/A | N/A | 2.23 | 106.21 | N/A | toxicant | N/A |
| ligand274 | N/A | N/A | 3.02 | 106.21 | N/A | toxicant | N/A |
| ligand230 | 1.18 | 0.69 | 2.18 | 2621.86 | 1013.52 | toxicant | Negative |
| ligand228 | 1.18 | 0.69 | 2.18 | 2621.86 | 1013.52 | toxicant | Negative |
| ligand223 | N/A | N/A | N/A | 396.53 | N/A | N/A | N/A |
| ligand36 | 1.40 | 1.05 | 3.20 | 2066.91 | 1239.14 | NON-toxicant | Negative |
| ligand112 | 2.74 | 0.52 | 12.29 | 984.94 | 173.28 | toxicant | Negative |
| ligand282 | 16.96 | 22.28 | 17.31 | 2215.19 | 60.73 | toxicant | Negative |

on the contrary, the binding affinity was used as a response variable along with other chemicals' descriptors as the predictor variables. A model with more than 87% predictive power was established based on four predictive variables, ATS1p, nCs, Hy, and F08[C-C]. The model was robust and predictive due to the unpredictable $Q^2$ statistics on decoys. No significant correlation was observed between predictors, except a positive correlation between ATS1p and F08[C-C]. This may suggest the impact of each variable on the predictive power of the model.

**Table 4. The QSAR summary of the developed models.** The fourth model with four predictive descriptors is reported in the present.

| Nbr. of variables | Variables | Variable IN/OUT | MSE | $R^2$ | Adjusted $R^2$ | Mallows' Cp | Akaike's AIC | Schwarz's SBC | Amemiya's PC |
|---|---|---|---|---|---|---|---|---|---|
| 1 | ATS1p | ATS1p | 18.0587 | 0.6649 | 0.6509 | | 73.3202 | 74.5390 | 0.3630 |
| 2 | ATS1p / F08[C-C] | F08[C-C] | 12.1815 | 0.7834 | 0.7645 | | 64.4134 | 66.8511 | 0.2543 |
| 3 | ATS1p / nCs / F08[C-C] | nCs | 8.2878 | 0.8590 | 0.8398 | | 55.6738 | 59.3304 | 0.1794 |
| 4 | ATS1p / nCs / Hy / F08[C-C] | Hy | 6.2156 | 0.8991 | 0.8799 | | 49.3177 | 54.1932 | 0.1394 |

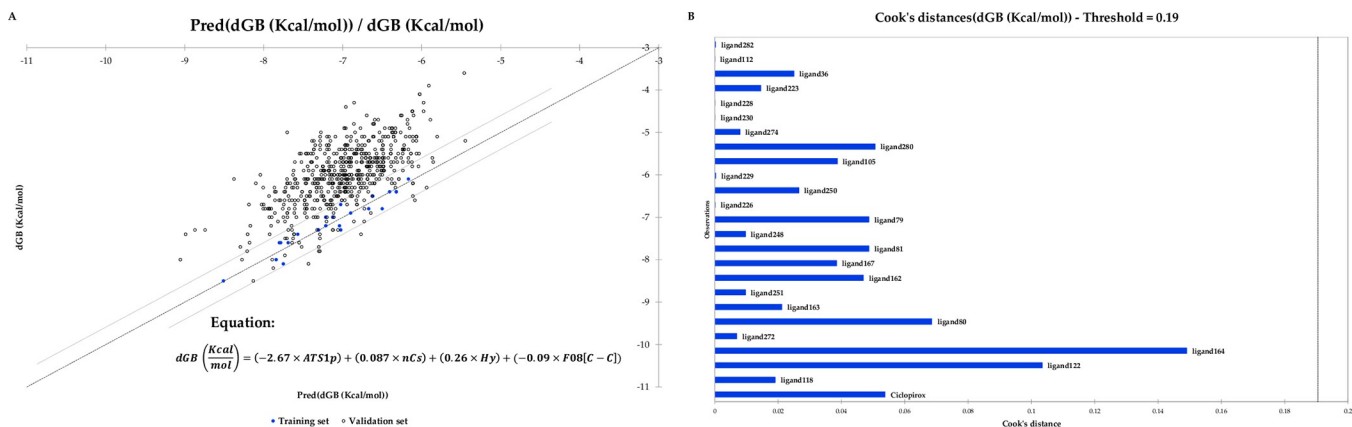

**Fig 3. The residual and Cook's distance plots.** A) shows a descent distribution of the training set ligands' residuals within upper and lower bounds with a 95% confidence interval (CI). B) Cook's distance values of the training set indicate no outliers, suggesting none of the ligands negatively affect the SAR model.

Furthermore, a robust statistical model was expected as the ligands are Ciclopirox derivatives. Because of the similar characteristics, it was assumed that the ligands might have the same binding site within the HBV core protein's hydrophobic pocket as reported for the Ciclopirox [25]. As mentioned above, Ciclopirox occupies a hydrophobic binding pocket within capsid CTD. Similarly, it was observed that the reconstructed ligands were located in a hydrophobic domain the same as Ciclopirox binding site. However, the constitution of amino acids in this site was extended.

Accordingly, hydrophobic phenylalanine residues (F23 and F24) were in close contact with the ligands. More importantly, F23 fully interacted with the ligands (please see the S1 Table). This implicates the critical role of F23 in capsid formation. Further, experimental studies can be done to estimate the role of F23 in capsid formation. Also, Y118 with 14 interactions and F122 with 13 interactions with Lig118 showed a significant role in stabilizing ligands within HBV capsid protein. These residues might be critical in sustaining dimer stabilization during capsid formation. It can be suggested that the Ciclopirox derivatives may offer the same

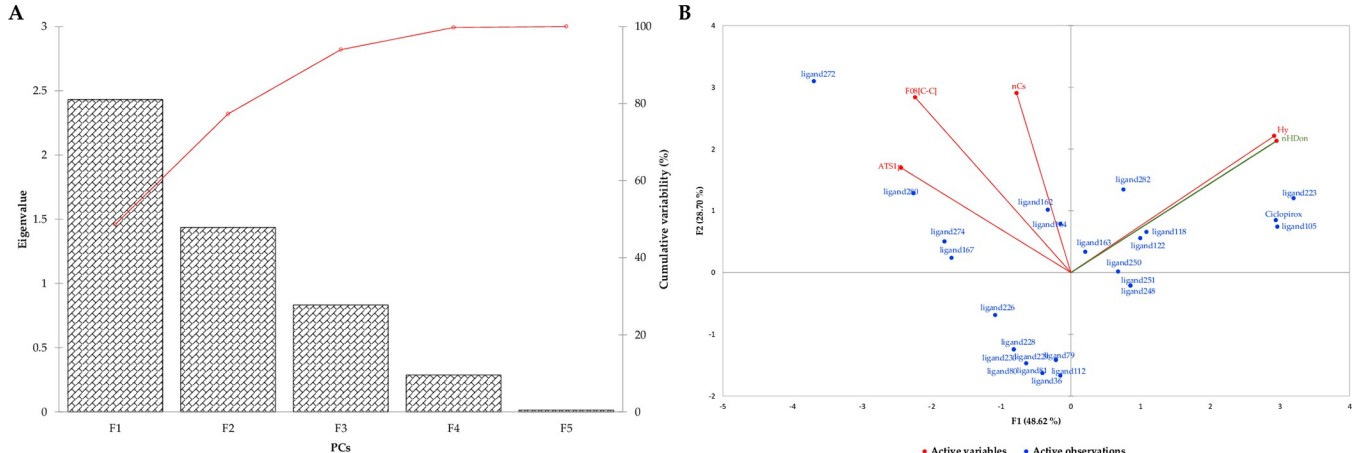

**Fig 4. The principal component analysis (PCA) of the QSAR model predictors in the training set.** A) The accumulative scree plot of four PCs with more than 77% variance in the $1^{st}$ two PCs. B) The biplot indicates the correlation of each predictor to eachothers. Accordingly, AST1p and F08[C-C] had the highest correlation toward PC2. It also shows the location of the loading factors of each ligand in the training set along with the PC factors. Data represents a higher effect of the F08[C-C] predictor on the model. nHDon represents the dummy variable.

mechanism of HBV replication inhibition due to the same physicochemical property and a property-matched similar binding site. It has been shown that Ciclopirox induces conformational changes and leads to asymmetrical capsid formation [25]. This could occur either by Ciclopirox binding to the capsid monomers or occupying a site where the dimers are formed. This is also supported by our results, indicating an insignificant change in the binding affinity of Ciclopirox to HBV core monomer or dimer (data are not shown).

In the presented study, a rational computation approach was used to reconstruct a set of potent anti-HBV compounds that may mimic the Ciclopirox activity to inhibit viral particle formation and subsequently reduce cccDNA level and DNA replication. The results allow future in vitro/vivo research on novel capsid inhibitors. Ciclopirox derivatives have been used to inhibit the replication of HSV-1 and HSV-2, the ligands introduced in this study can also be used to evaluate their biological activity against HSV types 1 and 2. Moreover, the protocol of this work can be employed for further drug discovery on either HBV or other clinically important viral diseases. Additionally, the molecular dynamic simulation (MDS) [33, 43] study can uncover the stability of highlighted compounds on induction of structural instability of HBV core and to further decipher the mechanism(s) of CAMs on malformed HBV capsid formation.

## Conclusion

The combination of computer-aided drug discovery and drug repurposing can pave the way to uncover the potential of FDA-approved drugs as antiviral agents. The present study's findings demonstrated that Ciclopirox derivatives might inhibit HBV virus assembly and subsequent viral replication inhibition by directly binding to the viral core protein carboxyl-terminal domain. Phe23 at the CTD and Tyr118 potentially play the most important role in viral capsid assembly, partially due to the highest interactions with the Ciclopirox derivatives. These ligands also have the same physicochemical properties that led to establishing a unique structure-activity relationship model.

## Supporting information

**S1 File. Raw data and results of the QSAR equation.** Data includes training sets with their respective predicted affinities (dependent variables) and 2D descriptors (independent variables).
(XLSX)

**S1 Table. The atom-atom interactions of Lig118 with the HBV capsid residues.**
(PDF)

## Acknowledgments

We thank the Department of Virology, School of Medicine, Iran University of Medical Sciences, Tehran, Iran for their spiritual support for this study.

## Author Contributions

**Conceptualization:** Alireza Mohebbi.

**Data curation:** Alireza Mohebbi, Shabnam Naderifar, Ali Salehnia Sammak.

**Formal analysis:** Touba Ghorbanzadeh, Shabnam Naderifar, Fatemeh Sana Askari.

**Investigation:** Touba Ghorbanzadeh.

**Methodology:** Alireza Mohebbi, Touba Ghorbanzadeh, Fattaneh Khalaj.

**Project administration:** Alireza Mohebbi.

**Resources:** Fatemeh Sana Askari.

**Software:** Alireza Mohebbi, Touba Ghorbanzadeh, Fatemeh Sana Askari.

**Supervision:** Alireza Mohebbi.

**Validation:** Alireza Mohebbi, Fattaneh Khalaj, Ali Salehnia Sammak.

**Visualization:** Alireza Mohebbi, Shabnam Naderifar.

**Writing – original draft:** Alireza Mohebbi, Touba Ghorbanzadeh, Shabnam Naderifar.

**Writing – review & editing:** Alireza Mohebbi, Shabnam Naderifar, Fattaneh Khalaj, Ali Salehnia Sammak.

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
