## [Decision Letter · Decision Letter 0]

31 Jan 2023

PONE-D-22-31671A Fragment-based drug discovery developed on Ciclopirox for inhibition of Hepatitis B virus core protein: An in silico studyPLOS ONE

Dear Dr. Mohebbi,

Thank you for submitting your manuscript to PLOS ONE. After careful consideration, we feel that it has merit but does not fully meet PLOS ONE’s publication criteria as it currently stands. Therefore, we invite you to submit a revised version of the manuscript that addresses the points raised during the review process.

We look forward to receiving your revised manuscript.

Kind regards,

Pramodkumar Pyarelal Gupta, PhD

Academic Editor

PLOS ONE

Journal Requirements:

Additional Editor Comments:

It seems the compound is not much big and very less fragment is possible hence the author should report the fragments, which will validate study.

What is mean ΔGB? And its significance.

Whether these all fragments are tested on same biological targets or some correlation?

Do the authors are working on QSAR model or SAR model?

In QSAR model what were the dependent and independent variables?

In discussion part the authors should discuss with the similar case.

Reviewers' comments:

Reviewer's Responses to Questions

**Comments to the Author**

1. Is the manuscript technically sound, and do the data support the conclusions?

Reviewer #1: Yes

2. Has the statistical analysis been performed appropriately and rigorously? 

Reviewer #1: Yes

3. Have the authors made all data underlying the findings in their manuscript fully available?

Reviewer #1: No

4. Is the manuscript presented in an intelligible fashion and written in standard English?

Reviewer #1: Yes

5. Review Comments to the Author

Reviewer #1: Interesting piece of work showing some new arenas for drug repurposing w.r.t Hepatitis B virus. However, I feel incorporation of a few changes might enhance the manuscript.

1. Interchanging paragraph 2 (49-57) and paragraph 3 (58-66):. Paragraph 1 and 3 talks about structure of the virus, while paragraph 2 and 4 talks about the drugs and treatments against the virus. Bringing paragraph 3 in place of paragraph 2 will make the introduction of Hepatitis B Virus seamless.

2. Not much has been written about Ciclopirox in the introduction and would suggest addition of a few more lines regarding Ciclopirox.

3. Comparison of the binding affinity of the target to a positive control will help in understanding of the efficacy of the drug fragment.

6. PLOS authors have the option to publish the peer review history of their article (what does this mean?). If published, this will include your full peer review and any attached files.

Reviewer #1: No

---

## [Author Response · Author response to Decision Letter 0]

9 Mar 2023

March 9, 2023

To: Editorial of PONE;

Thanks to the reviewer and editorials for their suggestions for improving the presentation of our paper, the comments and rephrases are presented in the revised manuscript. We hope that the revised version of the manuscript be more strengthened in details. Here, the comments of the reviewers and the editorial are addressed and also revised in the text.

1. Journal Requirements:

Authors response and action: The MS revised according to the template file. The Data Availability statement is updated by addition of the raw data file including all used information and statistics within the study. The figures were also revised in PACE server.

2. Additional Editor Comments:

1. It seems the compound is not much big and very less fragment is possible hence the author should report the fragments, which will validate study.

Author response and action: You are right, the derivatives are very small. We have revised the MS to describe the cause of smallness of the derivatives. Since the crystallographic structure of HBcAg complexed with Ciclopirox was used for FBDD, the resulted derivatives in the same binding site were small due to addition of fewer potent fragments inferring more suitable ∆GB affinities (Discussion section lines 208-210).

2. What is mean ΔGB? And its significance.

Author response and action: ΔGB was the free energy of binding, and we have revised it in the MS in the Fragment-based drug discovery sub-heading in M&M (lines 106-106). We have used the values of ΔGB for ranking the ligands. 

3. Whether these all fragments are tested on same biological targets or some correlation?

Author response and action: Thank you so much for the comment. We have only tested the fragments on a same target (HBcAg) in silico. 

4. Do the authors are working on QSAR model or SAR model?

Author response and action: It is working on QSAR model. Therefore, we have revised it through the MS.

5. In QSAR model what were the dependent and independent variables?

Author response and action: The affinity of the training and test sets were used as the dependent variable, and the chemical descriptors were set as the independent variables. This also mentioned and revised in the M&M section, subheading “Quantitative structure affinity relationship (QSAR)”

6. In discussion part the authors should discuss with the similar case.

Author response and action: We have discussed a similar paper in which the researchers developed Ciclopirox analogs for inhibiting the replication of human Herpes viruses (Lines 204-206).

3. Reviewers' comments:

Reviewer #1: Interesting piece of work showing some new arenas for drug repurposing w.r.t Hepatitis B virus. However, I feel incorporation of a few changes might enhance the manuscript.

1. Interchanging paragraph 2 (49-57) and paragraph 3 (58-66):. Paragraph 1 and 3 talks about structure of the virus, while paragraph 2 and 4 talks about the drugs and treatments against the virus. Bringing paragraph 3 in place of paragraph 2 will make the introduction of Hepatitis B Virus seamless.

Author response and action: AS recommended by dear review, P3 brought prior to the P2 in the revised MS.

2. Not much has been written about Ciclopirox in the introduction and would suggest addition of a few more lines regarding Ciclopirox.

Author response and action: As correctly recommended by the reviewer, more sentences is now added to the MS describing more about Ciclopirox. The revision lies in the last paragraph of the Introduction section (lines 85-88). 

3. Comparison of the binding affinity of the target to a positive control will help in understanding of the efficacy of the drug fragment.

Author response and action: Thank you so much for the comment. Ciclopirox was used as the positive control with the same binding site and also its affinity to the HBcAg was chosen as the cut-off value for selecting efficient ligands. This is revised in the first paragraph, and highlighted in yellow. 

Again, we are very grateful for considering our manuscript for peer-review. We hope the current version of the manuscript considered for publication in PONE.

Kind regards

Alireza Mohebbi

Ph.D. candidate in Medical Virology,

Department of Virology, School of Medicine,

Iran University of Medical Sciences, Tehran, Iran

E-mail: Mohebbi-a@goums.ac.ir

Tel: +98 9354674593

---

## [Decision Letter · Decision Letter 1]

5 May 2023

A Fragment-based drug discovery developed on Ciclopirox for inhibition of Hepatitis B virus core protein: An in silico study

PONE-D-22-31671R1

Dear Dr. Mohebbi,

We’re pleased to inform you that your manuscript has been judged scientifically suitable for publication and will be formally accepted for publication once it meets all outstanding technical requirements.

Kind regards,

Pramodkumar Pyarelal Gupta, PhD

Academic Editor

PLOS ONE

Additional Editor Comments (optional):

NA

Reviewers' comments:

Reviewer's Responses to Questions

**Comments to the Author**

1. If the authors have adequately addressed your comments raised in a previous round of review and you feel that this manuscript is now acceptable for publication, you may indicate that here to bypass the “Comments to the Author” section, enter your conflict of interest statement in the “Confidential to Editor” section, and submit your "Accept" recommendation.

Reviewer #1: All comments have been addressed

2. Is the manuscript technically sound, and do the data support the conclusions?

Reviewer #1: Yes

3. Has the statistical analysis been performed appropriately and rigorously? 

Reviewer #1: Yes

4. Have the authors made all data underlying the findings in their manuscript fully available?

Reviewer #1: Yes

5. Is the manuscript presented in an intelligible fashion and written in standard English?

Reviewer #1: Yes

6. Review Comments to the Author

Reviewer #1: (No Response)

7. PLOS authors have the option to publish the peer review history of their article (what does this mean?). If published, this will include your full peer review and any attached files.

Reviewer #1: No

---

## [Editor Report · Acceptance letter]

8 May 2023

PONE-D-22-31671R1 

A fragment-based drug discovery developed on ciclopirox for inhibition of Hepatitis B virus core protein: An in silico study 

Dear Dr. Mohebbi:

I'm pleased to inform you that your manuscript has been deemed suitable for publication in PLOS ONE. Congratulations! Your manuscript is now with our production department. 

Kind regards, 

on behalf of

Dr. Pramodkumar Pyarelal Gupta 

Academic Editor

PLOS ONE